# LatentStealth: Unnoticeable and Efficient Adversarial Attacks on Expressive Human Pose and Shape Estimation

## Abstract

Expressive human pose and shape estimation (EHPS) plays a central role in digital human generation, particularly in live-streaming applications. However, most existing EHPS models focus primarily on minimizing estimation errors, with limited attention on potential security vulnerabilities, such as generating inappropriate content, violent actions, or racially offensive gestures and expressions. Current adversarial attacks on EHPS models often generate visually conspicuous perturbations, limiting their practicality and ability to expose real-world security threats. To address this limitation, we propose an unnoticeable adversarial method, termed **LatentStealth**, specifically tailored for EHPS models. The key idea is to exploit the structured latent representations of natural images as the medium for crafting perturbations. Instead of injecting noise directly into the pixel space, our method projects inputs into the latent space, where adversarial patterns are generated and progressively refined along optimized directions. This latent-space manipulation enables the attack to maintain high imperceptibility while preserving its effectiveness. Furthermore, as the optimization process is guided by only a small number of model output queries, the framework achieves competitive attack performance with low computational overhead, making it both practical and efficient for real-world scenarios. Extensive experiments on the 3DPW and UBody datasets demonstrate the superiority of LatentStealth, revealing critical vulnerabilities in current systems. These findings highlight the urgent need to address and mitigate security risks in digital human generation technologies.

## 1 Introduction

Expressive human pose and shape estimation (EHPS) from monocular images or videos is fundamental to digital human modeling and key applications such as live streaming, virtual reality, and interactive gaming Loper et al. (2015); Pavlakos et al. (2019); Zhou et al. (2021); Zhang et al. (2023); Cai et al. (2024); Hong et al. (2022; 2021). In these scenarios, EHPS systems play a critical role in animating realistic digital avatars capable of dynamically replicating human movements and facial expressions in real time. Although recent advancements have significantly enhanced estimation accuracy, the security vulnerabilities of such models remain largely underexplored. Growing evidence Wang et al. (2021); Chen et al. (2024a); Schmalfuss et al. (2023); Zhou et al. (2024) indicates that these systems are susceptible to adversarial perturbations. Although current attack methods have demonstrated promising effectiveness, their perturbations are typically visible and therefore easily detected, which has limited attention to this issue within the research community. However, if attacks are conducted in an imperceptible manner, they may undermine public safety. For instance, in digital human live streaming, such attacks could cause an avatar's head to appear severed or clothing to fall off, leading to severe incidents involving horror or pornography. Therefore, raising awareness of this problem within both the research community and broader society is urgent.

While these models exhibit strong performance, their robustness against adversarial attacks has received limited attention. From a methodological standpoint, existing EHPS studies primarily focus on reducing errors in body pose estimation Zhou et al. (2021); Zhang et al. (2023) and facial pose estimation Jin et al. (2020); Moon et al. (2022); Lin et al. (2023); Cai et al. (2024). This is typically achieved using parametric human body models such as SMPL-X Pavlakos et al. (2019) or SMPLer-X

Cai et al. (2024), which leverage deep neural networks to capture the complex geometry and motion of the human body. TBA Li et al. (2025) was the first to propose a white-box transferable adversarial attack against EHPS. It utilizes the gradient of the SMPLer-X model to generate adversarial examples that significantly increase the estimation error across various EHPS models, thereby exposing potential security vulnerabilities. However, the perturbations produced by TBA are visually perceptible and fail to pass human inspection, which limits its applicability in real-world scenarios. Despite its effectiveness, such attacks have yet to attract sufficient attention from the research community. Considering these factors, we argue that an ideal adversarial attack on EHPS should meet three key criteria: **i)** the perturbation must be visually imperceptible; **ii)** the computational cost of the attacker should be minimal, aligning with realistic threat models; **iii)** the adversarial examples must achieve a high error growth rate. A substantial gap remains between current methods and these ideal standards.

To tackle this challenge, we propose a novel unnoticeable attack method, termed **LatentStealth**, against EHPS models. Inspired by Creswell et al. (2017); Salmona et al. (2022), LatentStealth perturbs the latent space of a pretrained Variational Autoencoder (VAE) Kingma et al. (2013); Ho & Salimans (2021) by injecting low-magnitude Gaussian noise. Leveraging the nonlinear generative capabilities of the decoder of VAE Khan & Storkey (2023); Upadhyay & Mukherjee (2021), our proposal can produce an initial adversarial example that is visually indistinguishable from the original input. The resulting pixel-wise difference is treated as an initial perturbation and subsequently refined through an iterative optimization process guided by a customized multi-task loss function. This loss simultaneously maximizes the discrepancy in the EHPS model's predictions while minimizing perceptual deviation from the clean image. Moreover, since the attack operates under a strict test budget, its computational cost remains low, ensuring efficiency. We conduct extensive experiments on the 3DPW and UBody datasets to evaluate the effectiveness of LatentStealth. Results show that it achieves strong attack capability (increasing pose estimation errors by an average of 17.27%–58.21%), high stealthiness (achieving the best FID Heusel et al. (2017) and PSNR Wang et al. (2004)), and low computational cost (requiring only three queries of the model outputs). In conclusion, our main contributions are:

• We propose the first unnoticeable adversarial attack method for EHPS systems under realistic constraints, which significantly degrades model performance while remaining visually undetectable. Moreover, our study highlights a critical yet underexplored security risk in digital human modeling.

• We develop **LatentStealth**, a VAE-based attack that perturbs latent representations with low-magnitude noise and leverages the decoder's nonlinear generative capacity to craft adversarial examples indistinguishable from clean inputs. This design fundamentally differs from pixel-space attacks by enhancing both stealthiness and naturalness.

• We design a multi-task loss that jointly balances adversarial effectiveness and perceptual similarity. Coupled with an iterative refinement process constrained by a strict test budget, LatentStealth achieves strong attack performance with low computational overhead, ensuring practical applicability in real-world scenarios.

## 2  RELATED WORK

**Expressive Human Pose and Shape Estimation (EHPS).** EHPS is fundamental to digital human modeling, with applications in virtual environments, live broadcasting, avatars, and animation Wang et al. (2020); Li et al. (2021); Pavlakos et al. (2018); Zou et al. (2021); Pang et al. (2022); Shen et al. (2023). Recent years have seen significant advances in improving the accuracy and realism of EHPS models. Pavlakos et al. (2019) introduced SMPL-X, a comprehensive 3D model incorporating articulated hands and expressive facial features for improved monocular pose capture. Moon et al. (2022) proposed Hand4Whole, leveraging metacarpophalangeal joints to predict wrist rotations precisely while excluding full-body context for better finger estimation. Lin et al. (2023) developed OSX, a one-stage framework using a component aware transformer to reconstruct whole-body meshes without post-processing. Cai et al. (2024) presented SMPLer-X, a generalist foundation model trained on diverse datasets, achieving strong generalization and competitive performance across EHPS benchmarks. However, existing research has predominantly concentrated on enhancing estimation accuracy, with limited attention to the models' vulnerability under attack threats.

**Adversarial Attack against Deep Neural Networks.** Adversarial attacks are commonly employed to assess the robustness of deep neural networks (DNNs) against malicious inputs Huang et al. (2021); Guo et al. (2019); Shen et al. (2023); Hu et al. (2022); Huang et al. (2020); Zhang et al. (2024). These attacks introduce subtle perturbations that mislead models while preserving the visual appearance of the input. Huang et al. (2024) proposed TT3D, which integrates NeRF-based multi-view reconstruction with dual optimization of texture features and MLP parameters to generate natural-looking, transferable 3D adversarial examples. Chen et al. (2023) presented the first content-aware, unrestricted adversarial attack based on diffusion models, enhancing transferability and realism by optimizing semantic perturbations in latent space. Chen et al. (2024b) proposed DiffAttack, a diffusion-based framework that achieves highly transferable and imperceptible attacks by jointly modulating cross-attention and self-attention mechanisms. Despite these advancements, the robustness of EHPS models remains largely underexplored. Li et al. (2025) proposed TBA consisting of a dual heterogeneous noise generator, the first EHPS-targeted attack; however, its visible perturbations limit its practicality. Consequently, TBA has failed to draw sufficient attention from the research community regarding EHPS security risks. To address this gap, we propose an unnoticeable attack method that reveals critical vulnerabilities in EHPS models.

## 3 PRELIMINARIES

**The Main Pipeline of EHPS.** Expressive human pose and shape estimation (EHPS) aims to reconstruct detailed 3D representations of the human body, face, and hands from monocular images. This capability underpins a wide array of applications, including virtual environments, live broadcasting, avatar generation, and animation. Formally, the ground-truth pose is denoted as $\alpha \in \mathbb{R}^{53 \times 3}$, comprising the body pose $\alpha_{\text{body}} \in \mathbb{R}^{22 \times 3}$, left- and right-hand poses $\alpha_{\text{lhand}}, \alpha_{\text{rhand}} \in \mathbb{R}^{15 \times 3}$, and the jaw pose $\alpha_{\text{jaw}} \in \mathbb{R}^{1 \times 3}$. Additionally, individual-specific geometry and facial expressions are encoded by the shape vector $\beta \in \mathbb{R}^{10}$ and the expression vector $\gamma \in \mathbb{R}^{10}$, respectively. Let $\mathcal{P}$ denote the EHPS model; given a unseen monocular image $x$, the objective is to estimate the complete set of pose, shape, and expression parameters $(\alpha, \beta, \gamma)$, e.g., $\mathcal{P}(x) = (\alpha, \beta, \gamma)$. The estimation process minimizes the parameter reconstruction loss:

$$\min_{\alpha, \beta, \gamma} \left( \|\alpha - \alpha_{gt}\|_2 + \|\beta - \beta_{gt}\|_2 + \|\gamma - \gamma_{gt}\|_2 \right). \tag{1}$$

Finally, 3D joints are rendered using the SMPL-X function $\mathcal{S}$ and a joint regressor $\mathcal{R}$, denoted as $\mathcal{R}(\mathcal{S}(\alpha, \beta, \gamma))$. $(\alpha_{gt}, \beta_{gt}, \gamma_{gt})$ denotes the ground-truth human pose and shape parameters.

**Threat Model.** This paper investigates adversarial attack techniques targeting EHPS models. Specifically, the attacker lacks access to any details of the model's training process, including the dataset, architecture, and trianing trick, and is only permitted a limited number of acquring the model's output. The generated adversarial examples are directly submitted to the model for inference. Other attack types with limited practical applicability (e.g., training-controlled or model modification attacks) require additional attacker capabilities and are beyond the scope of this study.

## 4 METHOD

### 4.1 PROBLEM FORMULATION

As noted in previous sections, existing EHPS models primarily aim to reduce human pose and shape estimation errors, with limited emphasis on their vulnerability to attacks. Although prior work Li et al. (2025) introduced the first adversarial attack against EHPS, exposing its inherent fragility, its real-world applicability remains constrained due to the visible nature of adversarial perturbations. This limitation hinders the community's full recognition of the intrinsic risks associated with EHPS. To tackle the challenge, we propose an unnoticeable attack method, termed LatentStealth, designed for realistic scenarios where the attacker can only access the model's output under a strict query budget.

**Attacker's Goals.** The LatentStealth is designed with two primary objectives: **1)** effectiveness and **2)** unnoticeability. Specifically, the effectiveness demands that the adversarial examples significantly increase errors in human pose and shape estimation, and the computational cost remains low. The

Figure 1: Overview of the LatentStealth pipeline. The framework consists of two stages: a) noise injection in latent space, and b) noise enhancement.

unnoticeability ensures that EHPS model owners are unable to identify the adversarial examples. The formalization is as follows:

$$\mathcal{P}(\hat{x}) = (\hat{\alpha}, \hat{\beta}, \hat{\gamma}), \quad \mathcal{P}(x) = (\alpha, \beta, \gamma), \tag{2}$$

$$\max_{\hat{\alpha}, \hat{\beta}, \hat{\gamma}} \left( \|\hat{\alpha} - \alpha\|_2^2 + \|\hat{\beta} - \beta\|_2^2 + \|\hat{\gamma} - \gamma\|_2^2 \right), \quad \text{s.t.} \quad \|\hat{x} - x\|_2^2 \leq \varepsilon, \tag{3}$$

where $x$ denotes a unseen monocular image (clean input image), $\hat{x}$ represents the corresponding adversarial sample image, and $\varepsilon$ indicates the perturbation threshold.

## 4.2 LATENTSTEALTH

Existing EHPS attacks often produce visible perturbations, limiting practicality. To overcome this, we propose **LatentStealth**, an unnoticeable attack method (as shown in Fig. 1) that perturbs the latent space of a pretrained Variational Autoencoder (VAE) Kingma et al. (2013) with low-magnitude noise and refines it via a multi-task loss under a strict test budget. LatentStealth achieves effective, efficient, and visually stealthy attacks, exposing critical yet overlooked security vulnerabilities in EHPS systems.

**Noise Injection in Latent Space.** Inspired by Shukla & Banerjee (2023), who questioned the necessity of norm-bounded pixel constraints, we introduce adversarial perturbations in the latent space, thereby more naturally preserving the structural features of the input and improving the imperceptibility of the perturbations. Unlike conventional methods that manipulate input images directly Goodfellow et al. (2014); Madry et al. (2017); Chen et al. (2023; 2024b), this strategy targets internal representations that more directly govern model predictions. Importantly, perturbing the latent space allows for adversarial behavior while preserving high perceptual fidelity in the image space. To reduce attack cost and improve real-world applicability, we employ a publicly available pretrained VAE Kingma et al. (2013) without additional training or fine-tuning.

The VAE is trained by maximizing the evidence lower bound and utilizes the reparameterization trick to enable differentiable sampling as follows:

$$\mathcal{L}(\phi, \theta; x) = \mathbb{E}_{q_\phi(z|x)}[\log p_\theta(x|z)] - D_{\mathrm{KL}}(q_\phi(z|x), p(z)), \tag{4}$$

$$z = \mu_\phi(x) + \sigma_\phi(x) \odot \epsilon_1, \quad \epsilon_1 \sim \mathcal{N}(0, I), \tag{5}$$

where $\phi$ and $\theta$ are the encoder and decoder parameters, the encoder outputs $\mu_\phi(x)$ and $\sigma_\phi(x)$ as mean and standard deviation, $z \sim \mathcal{N}(0, I)$ is the standard Gaussian prior, and the decoder $\mathcal{G}_\theta : \mathbb{R}^d \to \mathbb{R}^n$ maps the $d$-dimensional latent space to the $n$-dimensional pixel space, yielding a smooth latent representation Lee et al. (2025).

Building on this property, we inject an additional Gaussian noise $\epsilon_2$ into the latent representation $z$, creating a variationally grounded sampling-based perturbation that preserves the Gaussianity of the distribution. Specifically, we define $\hat{z} = z + \eta \cdot \epsilon_2, \epsilon_2 \sim \mathcal{N}(0, I)$, where $\eta$ is a hyperparameter controlling the perturbation magnitude, so that $\hat{z} \sim \mathcal{N}(0, (1 + \eta^2)I)$. When decoded by the decoder $\mathcal{G}_\theta$, the perturbed latent representation $\hat{z}$ yields outputs $\hat{x} = \mathcal{G}_\theta(\hat{z})$. We perform a first-order Taylor expansion of $\mathcal{G}_\theta(z)$:

$$\mathcal{G}_\theta(z + \eta \cdot \epsilon_2) = \mathcal{G}_\theta(z) + J_{\mathcal{G}_\theta}(z) \cdot (\eta \cdot \epsilon_2) + o(\|\eta \cdot \epsilon_2\|_2), \tag{6}$$

where $J_{\mathcal{G}_\theta}(z)$ denotes the Jacobian of the decoder $\mathcal{G}_\theta$ at $z$. To ensure the first-order approximation remains valid, we restrict $\eta = 0.05$, so that $\eta \cdot \epsilon_2$ is sufficiently small compared to $J_{\mathcal{G}_\theta}(z) \cdot (\eta \cdot \epsilon_2)$. As a result, we can omit the higher-order term $o(||\eta \cdot \epsilon_2||_2)$:

$$\hat{x} \approx \mathcal{G}_\theta(z) + J_{\mathcal{G}_\theta}(z) \cdot (\eta \cdot \epsilon_2). \tag{7}$$

Consequently, the perturbation $\Delta x = \hat{x} - x$ in the pixel space can be expressed as:

$$\Delta x \approx J_{\mathcal{G}_\theta}(z) \cdot (\eta \cdot \epsilon_2). \tag{8}$$

Specifically, the perturbation $\Delta x$ in the pixel space exhibits a linear relationship with the perturbation $\eta \cdot \epsilon_2$ in the latent space. Since the pre-trained model $\mathcal{G}_\theta$ is typically continuously differentiable, the magnitude of the perturbation is consequently constrained as follows:

$$\Delta x \approx J_{\mathcal{G}_\theta}(z) \cdot (\eta \cdot \epsilon_2) \leq \eta \cdot ||J_{\mathcal{G}_\theta}(z)||_2 \cdot ||\epsilon_2||_2. \tag{9}$$

If the perturbation magnitude $\eta$ is sufficiently small, the corresponding distortion in pixel space remains imperceptible, even in directions associated with large Jacobian values.

Similarly, a first-order approximation is applied to the target model:

$$\mathcal{P}(\hat{x}) = \mathcal{P}(x + J_{\mathcal{G}_\theta}(z) \cdot (\eta \cdot \epsilon_2)) \approx \mathcal{P}(x) + J_{\mathcal{P}}(x) \cdot J_{\mathcal{G}_\theta}(z) \cdot (\eta \cdot \epsilon_2). \tag{10}$$

We define the threshold for significant changes in the estimation parameters of the EHPS model as $\delta$. If the deviation exceeds this threshold, the model is considered to be misled, i.e., $||\mathcal{P}(\hat{x}) - \mathcal{P}(x)||_2 \geq \delta$. According to Eq. 10, this leads to the inequality $J_{\mathcal{P}}(x) \cdot J_{\mathcal{G}_\theta}(z) \cdot (\eta \cdot \epsilon_2) \geq \delta$.

By combining the Jacobian matrices into a single term $\mathbb{J} = J_{\mathcal{P}}(x) \cdot J_{\mathcal{G}_\theta}(z)$, the inequality can be reformulated as $||\mathbb{J} \cdot (\eta \cdot \epsilon_2)||_2 \geq \delta$. Given that $\epsilon_2 \sim \mathcal{N}(0, I)$, the expected perturbation is:

$$\mathbb{E}[||\mathbb{J} \cdot (\eta \cdot \epsilon_2)||_2^2] = \eta^2 \mathbb{E}[\epsilon_2^\top \mathbb{J}^\top \mathbb{J} \epsilon_2] = \eta^2 \mathrm{tr}(\mathbb{J}^\top \mathbb{J}). \tag{11}$$

To minimize the perturbation magnitude $\eta$ while effectively crossing the decision boundary, the perturbation should be aligned with the right singular vector corresponding to the largest singular value:

$$\mathbb{J} = U\Sigma V^\top, \quad \Sigma = \mathrm{diag}(\sigma_1, \sigma_2, \ldots), \quad \sigma_1 \geq \sigma_2 \geq \cdots \tag{12}$$

The most sensitive direction corresponds to the first column vector $v_1$ of $V$. If $\epsilon_2$ is set to $v_1$, the condition for the minimum perturbation magnitude becomes:

$$||\mathbb{J} \cdot (\eta \cdot v_1)||_2 = \eta \cdot \sigma_1 \geq \delta \Rightarrow \eta \geq \frac{\delta}{\sigma_1}. \tag{13}$$

In summary, an attacker can amplify variations in sensitive features by injecting small perturbations into the latent space, causing the prediction to deviate rapidly from its original stable state and leading to significant errors.

**Noise Enhancement.** Inspired by Projected Gradient Descent (PGD) Madry et al. (2017), we adopt an iterative refinement strategy to improve the effectiveness of adversarial examples, guided by feedback from the EHPS model. However, updating the noise in the latent space entails significant computational overhead. Specifically, the latent perturbation is defined as:

$$\mathcal{P}(\mathcal{G}_\theta(z + \Delta \hat{z}^{(t)})) = (\hat{\alpha}^{(t)}, \hat{\beta}^{(t)}, \hat{\gamma}^{(t)}), \quad \mathcal{P}(x) = (\alpha, \beta, \gamma), \tag{14}$$

$$\Delta \hat{z}^{(t+1)} = \Delta \hat{z}^{(t)} + \xi_z \nabla_{\Delta \hat{z}^{(t)}} \mathcal{L}((\hat{\alpha}^{(t)}, \hat{\beta}^{(t)}, \hat{\gamma}^{(t)}), (\alpha, \beta, \gamma)), \tag{15}$$

$$\nabla_{\Delta \hat{z}^{(t)}} \mathcal{L}((\hat{\alpha}^{(t)}, \hat{\beta}^{(t)}, \hat{\gamma}^{(t)}), (\alpha, \beta, \gamma)) = \frac{\partial \mathcal{L}}{\partial \mathcal{G}_\theta} \frac{\partial \mathcal{G}_\theta}{\partial \Delta \hat{z}^{(t)}}, \tag{16}$$

where $t$ denotes the iteration step, $\Delta \hat{z}^{(t)}$ represent the latent perturbation and its update at step $t$, $z$ is the latent representation of clean image, $(\alpha, \beta, \gamma)$ are the outputs obtained by feeding the clean image into $\mathcal{P}$, and $\xi_z$ is the learning rate for the gradient update in the latent space. The term $\frac{\partial \mathcal{G}_\theta}{\partial \Delta \hat{z}^{(t)}}$ denotes the Jacobian matrix of the decoder, whose computation is prohibitively expensive due to its high dimensionality ($n \times d$, often very large). Furthermore, each update necessitates a complete forward pass through the decoder network, resulting in substantial memory and computational consumption. Consequently, performing iterative updates in the latent space is highly inefficient (see Table 2).

To address the aforementioned issue, we propose enhancing the noise directly in the pixel space. The initial pixel-space perturbation is defined as:

$$\Delta x^{(0)} = x^{(0)} - x, \quad x^{(0)} = \mathcal{G}_\theta(z + \eta \cdot \epsilon_2). \tag{17}$$

We perform iterative updates to the perturbation in pixel space as follows:

$$\mathcal{P}(x + \Delta x^{(t)}) = (\hat{\alpha}^{(t)}, \hat{\beta}^{(t)}, \hat{\gamma}^{(t)}), \tag{18}$$

$$\Delta x^{(t+1)} = \Delta x^{(t)} + \xi_{\Delta x} \cdot \nabla_{\Delta x^{(t)}} \mathcal{L}((\hat{\alpha}^{(t)}, \hat{\beta}^{(t)}, \hat{\gamma}^{(t)}), (\alpha, \beta, \gamma)), \tag{19}$$

$$\nabla_{\Delta x^{(t)}} \mathcal{L}((\hat{\alpha}^{(t)}, \hat{\beta}^{(t)}, \hat{\gamma}^{(t)}), (\alpha, \beta, \gamma)) = \frac{\partial \mathcal{L}}{\partial(x + \Delta x^{(t)})}, \tag{20}$$

where $\xi_{\Delta x}$ denotes the learning rate for gradient updates in pixel space. Importantly, computing gradients in pixel space does not require backpropagation through the decoder network. Instead, perturbations are applied directly to the original image $x$, thereby eliminating the need to compute the high-dimensional Jacobian transformation from latent to pixel space. This approach substantially reduces both the computational complexity and memory usage of each gradient update step.

Furthermore, to ensure strong adversarial strength while maintaining the perturbation's imperceptibility, we incorporate a regularization term into the optimization process. To this end, we introduce a novel multi-task loss function to jointly achieve these objectives:

$$\mathcal{L} = -\mathbb{E}_{\Delta x} \left[ \|\hat{\alpha}^{(t)} - \alpha\|_2^2 + \|\hat{\beta}^{(t)} - \beta\|_2^2 + \|\hat{\gamma}^{(t)} - \gamma\|_2^2 \right] + \frac{1}{w \times h} \sum_{i=0}^{w-1} \sum_{j=0}^{h-1} \left\| \Delta x^{(t)}(i,j) \right\|_2^2, \tag{21}$$

where $w$ and $h$ denote the width and height of $x$, respectively. The first term encourages the perturbed pose parameters to diverge from those of $\mathcal{P}(x)$, thereby enhancing the effectiveness of the attack. The second term imposes a strict constraint on pixel-level variations to ensure the imperceptibility of the perturbation, effectively serving as a regularization term. According the Eqs 18, 20 and 21, the update rule for the pixel-space perturbation is formulated as:

$$\Delta x^{(t+1)} = \Delta x^{(t)} + \xi_{\Delta x} \cdot \left( \frac{\partial \mathcal{L}}{\partial(x + \Delta x^{(t)})} - \lambda \cdot x^{(t)} \right), \tag{22}$$

where $\lambda$ is a hyperparameter that balances attack strength and imperceptibility. It is evident that the computational complexity of updates in latent space is $\mathcal{O}(n \times d)$, whereas the complexity of pixel-space updates is only $\mathcal{O}(n)$. Therefore, the proposed noise enhancement strategy in pixel space incurs significantly lower computational overhead (see the Method C and **LatentStealth** in Table 2).

## 5 EXPERIMENTS

### 5.1 EXPERIMENTAL SETTINGS

**Datasets and EHPS Models.** We evaluate our method on two benchmarks: 3DPW Von Marcard et al. (2018), with in-the-wild RGB videos and 3D SMPL pose annotations, and UBody Lin et al. (2023), a large-scale upper-body mesh dataset with 1.05 million annotated images across 15 scenarios. For evaluation, we test LatentStealth on the three most representative EHPS models, including Hand4Whole Moon et al. (2022), OSX Lin et al. (2023), and SMPLer-X Cai et al. (2024), which provides four ViT-based pre-trained variants, denoted as SMPLer-X-M, where M denotes the size of the Vision Transformer (ViT) backbone Alexey (2020) (M ∈ S, B, L, H).

**Baseline and Evaluation Metrics.** Currently, only one adversarial attack targets EHPS. To evaluate LatentStealth, we compare it with state-of-the-art methods in computer vision, including classical attacks (FGSM Goodfellow et al. (2014), PGD Madry et al. (2017)), diffusion-based approaches (ACA Chen et al. (2023), DiffAttack Chen et al. (2024b)), and the EHPS-specific TBA Li et al. (2025). We report MPJPE and MPVPE for 3D joint and mesh accuracy, and PA-MPJPE/PA-MPVPE for alignment-aware evaluation, following prior EHPS benchmarks Moon et al. (2022); Lin et al. (2023); Cai et al. (2024). All metrics are in millimeters (mm).

**Implementation Details.** The experiments are conducted on a system running Ubuntu 20.04.6 LTS, using PyTorch framework, with a single NVIDIA L20 GPU. The VAE model employed is

Table 1: Performance comparison of different adversarial attack methods on state-of-the-art EHPS models on the UBody dataset. The error growth rates are marked in gray. The maximum error and maximum error growth rate on each setting are **highlighted** underlined.

| Model | Attack | PA MPVPE ↓ (mm) | | | MPVPE ↓ (mm) | | | PA MPJPE ↓ (mm) | |
|---|---|---|---|---|---|---|---|---|---|
| | | All | Hands | Face | All | Hands | Face | Body | Hands |
| SMPLer-X-H | Clean | 24.64 | 8.47 | 2.24 | 41.22 | 30.88 | 16.62 | 29.29 | 8.64 |
| | FGSM | 29.83 (21.06%) | 9.03 (6.61%) | 2.41 (7.59%) | 57.76 (40.13%) | 36.03 (16.68%) | 20.11 (21.00%) | 34.82 (18.88%) | 9.18 (6.25%) |
| | PGD | 29.59 (20.09%) | 9.17 (8.26%) | 2.49 (11.16%) | 60.06 (45.71%) | 34.50 (11.72%) | 22.00 (32.37%) | 34.55 (17.96%) | 9.33 (7.99%) |
| | ACA | 26.84 (8.93%) | 9.14 (7.91%) | 2.40 (7.14%) | 46.29 (12.30%) | 34.12 (10.49%) | 18.04 (8.54%) | 31.39 (7.17%) | 9.32 (7.87%) |
| | DiffAttack | 28.54 (15.83%) | 9.32 (10.04%) | 2.43 (8.48%) | 53.41 (29.57%) | 35.41 (14.67%) | 19.70 (18.53%) | 33.80 (15.40%) | 9.48 (9.72%) |
| | TBA | 30.00 (21.75%) | 9.09 (7.32%) | 2.43 (8.48%) | 58.13 (41.02%) | 36.29 (17.52%) | 21.19 (27.50%) | 35.36 (20.72%) | 9.25 (7.06%) |
| | LatentStealth (Ours) | **43.94** (78.33%) | **9.90** (16.88%) | **2.80** (25.00%) | **88.60** (114.94%) | **48.69** (57.69%) | **28.38** (70.76%) | **50.48** (72.35%) | **10.08** (16.67%) |
| SMPLer-X-L | Clean | 25.66 | 8.98 | 2.39 | 43.26 | 33.11 | 17.54 | 30.14 | 9.16 |
| | FGSM | 32.88 (28.14%) | 9.32 (3.79%) | 2.51 (5.02%) | 63.99 (47.92%) | 39.44 (19.12%) | 22.52 (28.39%) | 38.58 (28.00%) | 9.49 (3.60%) |
| | PGD | 30.82 (20.11%) | 9.36 (4.23%) | 2.62 (9.62%) | 63.07 (45.79%) | 36.61 (10.57%) | 22.84 (30.22%) | 36.40 (20.77%) | 9.52 (3.93%) |
| | ACA | 28.13 (9.63%) | 9.69 (7.91%) | 2.52 (5.44%) | 49.39 (14.17%) | 36.47 (10.15%) | 19.18 (9.35%) | 32.50 (7.83%) | 9.86 (7.64%) |
| | DiffAttack | 30.65 (19.45%) | 9.74 (8.46%) | 2.49 (4.18%) | 57.57 (33.08%) | 37.96 (14.65%) | 20.53 (17.05%) | 36.32 (20.50%) | 9.90 (8.08%) |
| | TBA | 33.50 (30.55%) | 9.43 (5.01%) | 2.69 (12.55%) | 66.16 (52.94%) | 40.50 (22.32%) | 22.05 (25.71%) | 39.01 (29.73%) | 9.60 (4.80%) |
| | LatentStealth (Ours) | **47.86** (86.52%) | **9.83** (9.47%) | **2.92** (22.18%) | **91.46** (114.42%) | **52.17** (57.57%) | **33.01** (88.20%) | **54.50** (80.82%) | **10.03** (9.50%) |
| SMPLer-X-B | Clean | 28.95 | 9.72 | 2.60 | 50.75 | 38.02 | 19.82 | 33.43 | 9.91 |
| | FGSM | 38.06 (31.47%) | 10.06 (3.50%) | 2.83 (8.85%) | 78.25 (54.19%) | 45.42 (19.46%) | 25.47 (28.51%) | 43.83 (31.11%) | 10.24 (3.33%) |
| | PGD | 34.56 (19.38%) | 9.85 (1.34%) | 2.87 (10.38%) | 71.73 (41.34%) | 41.46 (9.05%) | 24.47 (23.46%) | 40.16 (20.13%) | 10.03 (1.21%) |
| | ACA | 31.48 (8.74%) | 10.11 (4.01%) | 2.74 (5.38%) | 57.02 (12.35%) | 41.31 (8.65%) | 21.99 (10.95%) | 36.16 (8.17%) | 10.30 (3.94%) |
| | DiffAttack | 32.04 (10.67%) | 10.12 (4.12%) | 2.72 (4.62%) | 58.23 (14.74%) | 41.48 (9.10%) | 21.79 (9.94%) | 37.18 (11.22%) | 10.30 (3.94%) |
| | TBA | 37.44 (29.33%) | 9.80 (0.82%) | 2.90 (11.54%) | 76.41 (50.56%) | 46.25 (21.65%) | 25.63 (29.31%) | 42.69 (27.70%) | 10.04 (1.31%) |
| | LatentStealth (Ours) | **53.38** (84.39%) | **10.61** (9.16%) | **3.23** (24.23%) | **106.97** (110.78%) | **57.10** (50.18%) | **36.21** (82.69%) | **62.12** (85.82%) | **10.82** (9.18%) |
| SMPLer-X-S | Clean | 32.28 | 10.05 | 2.83 | 57.23 | 42.80 | 22.26 | 37.38 | 10.27 |
| | FGSM | 40.02 (23.98%) | 10.73 (6.77%) | 3.01 (6.36%) | 82.44 (44.05%) | 49.33 (15.26%) | 29.30 (31.63%) | 46.44 (24.24%) | 10.96 (6.72%) |
| | PGD | 36.90 (14.31%) | 10.63 (5.77%) | 3.07 (8.48%) | 78.03 (36.43%) | 45.79 (6.99%) | 26.85 (20.62%) | 43.37 (16.02%) | 10.89 (6.04%) |
| | ACA | 34.48 (6.82%) | 10.53 (4.78%) | 2.93 (3.53%) | 61.99 (8.32%) | 45.04 (5.23%) | 23.63 (6.15%) | 39.66 (6.10%) | 10.72 (4.38%) |
| | DiffAttack | 34.27 (6.16%) | 10.54 (4.88%) | 2.89 (2.12%) | 61.31 (7.13%) | 45.25 (5.72%) | 23.47 (5.44%) | 39.43 (5.48%) | 10.74 (4.58%) |
| | TBA | 39.52 (22.43%) | 10.96 (9.05%) | 3.17 (12.01%) | 81.48 (42.37%) | 49.46 (15.56%) | 28.72 (29.02%) | 46.06 (23.22%) | **11.25** (9.54%) |
| | LatentStealth (Ours) | **50.16** (55.39%) | 10.53 (4.78%) | **3.33** (17.67%) | **98.24** (71.66%) | **59.06** (37.99%) | **34.70** (55.88%) | **58.62** (56.82%) | 10.78 (4.97%) |
| OSX | Clean | 40.85 | 9.37 | 3.54 | 86.36 | 42.72 | 25.72 | 49.26 | 9.62 |
| | FGSM | 45.68 (11.82%) | 9.86 (5.23%) | 3.29 (-7.06%) | 104.51 (21.02%) | 46.71 (9.34%) | 33.64 (30.84%) | 54.17 (9.97%) | 10.13 (5.30%) |
| | PGD | 42.47 (3.97%) | 9.68 (3.31%) | 3.00 (-15.25%) | 97.86 (13.32%) | 43.89 (2.74%) | 28.78 (11.94%) | 51.11 (3.76%) | 9.95 (3.43%) |
| | ACA | 41.93 (2.64%) | 9.66 (3.09%) | 3.48 (-1.69%) | 89.10 (3.17%) | 45.02 (5.38%) | 27.05 (5.21%) | 49.92 (1.34%) | 9.94 (3.33%) |
| | DiffAttack | 40.66 (-0.47%) | 9.67 (3.20%) | 3.11 (-12.15%) | 88.86 (2.89%) | 43.99 (2.97%) | 26.04 (1.28%) | 48.64 (-1.26%) | 9.95 (3.43%) |
| | TBA | 41.37 (1.27%) | 9.67 (3.20%) | 2.98 (-15.82%) | 97.25 (12.61%) | 45.37 (6.20%) | 28.05 (9.10%) | 49.57 (0.63%) | 9.94 (3.33%) |
| | LatentStealth (Ours) | **60.36** (47.76%) | **10.44** (11.42%) | **3.86** (9.04%) | **134.39** (55.62%) | **55.82** (30.66%) | **47.73** (85.65%) | **75.17** (52.60%) | **10.73** (11.54%) |
| Hand4Whole | Clean | 41.04 | 7.85 | 2.96 | 93.81 | 35.60 | 30.30 | 49.80 | 7.97 |
| | FGSM | 41.03 (-0.02%) | 7.85 (0.00%) | 2.96 (0.00%) | 93.82 (0.01%) | 35.61 (0.03%) | 30.30 (0.00%) | 49.80 (0.00%) | 7.97 (0.00%) |
| | PGD | 41.06 (0.05%) | 7.85 (0.00%) | 2.96 (0.00%) | 93.87 (0.06%) | 35.62 (0.06%) | 30.31 (0.03%) | 49.82 (0.04%) | 7.97 (0.00%) |
| | ACA | 41.40 (0.88%) | 7.83 (-0.25%) | 2.98 (0.68%) | 95.15 (1.43%) | 36.02 (1.18%) | 30.71 (1.35%) | 50.34 (1.08%) | 7.95 (-0.25%) |
| | DiffAttack | 41.10 (0.15%) | 7.99 (1.78%) | 2.91 (-1.69%) | 94.79 (1.04%) | 36.02 (1.18%) | 35.11 (15.87%) | 50.12 (0.64%) | 8.10 (1.63%) |
| | TBA | 43.72 (6.53%) | 8.10 (3.18%) | 3.00 (1.35%) | 104.31 (11.19%) | 36.57 (2.72%) | 34.52 (13.93%) | 53.22 (6.87%) | 8.22 (3.14%) |
| | LatentStealth (Ours) | **56.59** (37.89%) | **9.74** (24.08%) | **3.52** (18.92%) | **128.57** (37.05%) | **54.02** (51.74%) | **41.83** (38.05%) | **71.67** (43.92%) | **9.93** (24.59%) |

*stabilityai/sd-vae-ft-mse* from Hugging Face [1]. The maximum perturbation magnitude is set to $\eta = 0.05$ with a step size of 0.01, and an $l_\infty$ constraint was enforced to ensure that changes in individual pixel values did not exceed 0.05. For baseline methods involving iterative optimization, the number of iterations is fixed at 20. For LatentStealth, the number of queries is set to $t = 3$, and the balancing hyperparameter is fixed at $\lambda = 0.5$.

## 5.2 ATTACK PERFORMANCE ON EHPS MODELS

We first evaluate the effectiveness of LatentStealth on the UBody dataset, which includes full-body, hand, and facial regions of digital humans. As shown in Table 1, across all tasks, the adversarial examples generated by LatentStealth cause the most significant degradation in the accuracy of all EHPS models, compared to other attack methods. Specifically, the average increase in estimation error ranged from 25.13% to 58.21%. Significantly, for the SMPLer-X-H model, the MPVPE (All) (sixth column) increased from 41.22 mm to 88.60 mm, corresponding to a 114.94% error growth rate.

We further evaluate LatentStealth on the 3DPW dataset, focusing on how adversarial examples impact the estimation of body parts across different EHPS models. As shown in Table 3, LatentStealth consistently causes the greatest reduction in model accuracy. Especially, LatentStealth yields a minimum error growth rate of 17.92% (increasing PA MPJPE from 67.93 mm to 80.10 mm on the Hand4Whole model) and a maximum of 48.46% (raising MPJPE from 79.46 mm to 118.00 mm on the SMPLer-X-B model), with average increases ranging from 17.27% to 44.49%.

These results demonstrate that LatentStealth is highly destructive. Remarkably, its computational cost remains low, ensuring efficiency. Interestingly, we observe negative error growth rates in certain cases, such as PA MPVPE (Face) under the OSX model on UBody and several other metrics under specific settings. This finding suggests that while adversarial attacks generally impair EHPS performance, they may, in rare instances, also enhance estimation accuracy.

---

[1]An open source platform. https://huggingface.co/stabilityai/sd-vae-ft-mse

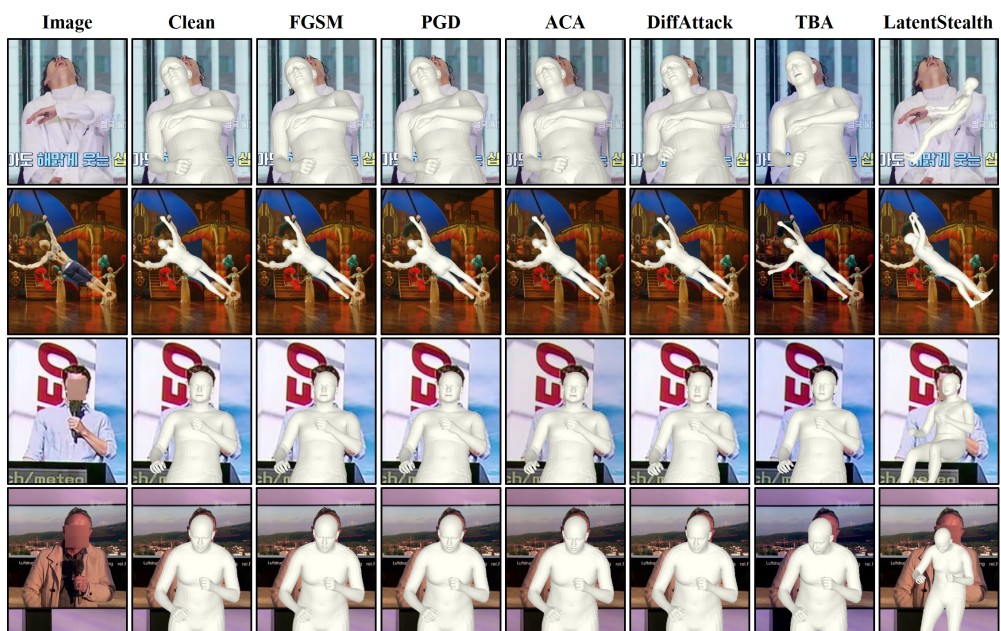

Figure 2: Visualizing various adversarial samples for digital human generation.

Table 2: Ablation study of different adversarial attack frameworks on state-of-the-art EHPS models on the UBody dataset. The error growth rates are marked in gray. The maximum error and maximum error growth rate on each setting are **highlighted** underlined.

| Model | Method | PA MPVPE ↓ (mm) | | | MPVPE ↓ (mm) | | | PA MPJPE ↓ (mm) | | Memory |
|---|---|---|---|---|---|---|---|---|---|---|
| | | All | Hands | Face | All | Hands | Face | Body | Hands | |
| SMPLer-X-H | Clean | 24.64 | 8.47 | 2.24 | 41.22 | 30.88 | 16.62 | 29.29 | 8.64 | - |
| | A | 32.41 (31.53%) | 8.98 (6.02%) | 2.41 (7.59%) | 63.43 (53.88%) | 38.65 (25.16%) | 20.16 (21.30%) | 38.29 (30.73%) | 9.12 (5.56%) | 9.93 GB |
| | B | 26.27 (6.62%) | 8.58 (1.30%) | 2.22 (-0.89%) | 47.68 (15.67%) | 32.28 (4.53%) | 18.07 (8.72%) | 31.27 (6.76%) | 8.74 (1.16%) | 7.00 GB |
| | C | | | | Out of Memory | | | | | ≥ 45 GB |
| | LatentStealth (Ours) | **43.94** (78.33%) | **9.90** (16.88%) | **2.80** (25.00%) | **88.60** (114.94%) | **48.69** (57.69%) | **28.38** (70.76%) | **50.48** (72.35%) | **10.08** (16.67%) | 12.83 GB |
| SMPLer-X-L | Clean | 25.66 | 8.98 | 2.39 | 43.26 | 33.11 | 17.54 | 30.14 | 9.16 | - |
| | A | 35.64 (38.89%) | 9.18 (2.23%) | 2.48 (3.77%) | 69.94 (61.97%) | 41.61 (25.67%) | 23.45 (33.69%) | 42.31 (40.38%) | 9.35 (2.07%) | 6.20 GB |
| | B | 27.53 (7.29%) | 8.99 (0.11%) | 2.42 (1.26%) | 50.45 (16.62%) | 34.09 (2.96%) | 19.09 (8.84%) | 32.31 (7.20%) | 9.16 (0.00%) | 6.77 GB |
| | C | 27.24 (6.16%) | 8.99 (0.11%) | 2.40 (0.42%) | 47.74 (10.36%) | 34.21 (3.32%) | 18.37 (4.73%) | 32.05 (6.34%) | 9.17 (0.11%) | 43.79 GB |
| | LatentStealth (Ours) | **47.86** (86.52%) | **9.83** (9.47%) | **2.92** (22.18%) | **91.46** (114.42%) | **52.17** (57.57%) | **33.01** (88.20%) | **54.50** (80.82%) | **10.03** (9.50%) | 8.25 GB |
| SMPLer-X-B | Clean | 28.95 | 9.72 | 2.60 | 50.75 | 38.02 | 19.82 | 33.43 | 9.91 | - |
| | A | 37.01 (27.84%) | 9.68 (-0.41%) | 2.71 (4.23%) | 74.09 (45.99%) | 44.58 (17.25%) | 24.02 (21.19%) | 43.11 (28.96%) | 9.86 (-0.50%) | 3.45 GB |
| | B | 31.48 (8.74%) | 9.49 (-2.37%) | 2.59 (-0.38%) | 56.79 (11.90%) | 39.01 (2.60%) | 21.56 (8.78%) | 37.00 (10.68%) | 9.67 (-2.42%) | 6.24 GB |
| | C | 27.24 (-5.91%) | 8.99 (-7.51%) | 2.40 (-7.69%) | 47.74 (-5.93%) | 34.21 (-10.02%) | 18.37 (-7.32%) | 32.05 (-4.13%) | 9.17 (-7.47%) | 43.63 GB |
| | LatentStealth (Ours) | **53.38** (84.39%) | **10.61** (9.16%) | **3.23** (24.23%) | **106.97** (110.78%) | **57.10** (50.18%) | **36.21** (82.69%) | **62.12** (85.82%) | **10.82** (9.18%) | 8.32 GB |
| SMPLer-X-S | Clean | 32.28 | 10.05 | 2.83 | 57.23 | 42.80 | 22.26 | 37.38 | 10.27 | - |
| | A | 38.80 (20.20%) | 10.38 (3.28%) | 2.85 (0.71%) | 76.07 (32.92%) | 48.19 (12.59%) | 25.81 (15.95%) | 45.44 (21.56%) | 10.60 (3.21%) | 2.73 GB |
| | B | 35.74 (10.72%) | 10.02 (-0.30%) | 2.88 (1.77%) | 67.46 (17.88%) | 44.61 (4.23%) | 23.70 (6.47%) | 41.41 (10.78%) | 10.26 (-0.10%) | 7.04 GB |
| | C | 33.89 (4.99%) | 10.10 (0.50%) | 2.85 (0.71%) | 60.28 (5.33%) | 44.20 (3.27%) | 22.74 (2.16%) | 39.21 (4.90%) | 10.34 (0.68%) | 43.33 GB |
| | LatentStealth (Ours) | **50.16** (55.39%) | **10.53** (4.78%) | **3.33** (17.67%) | **98.24** (71.66%) | **59.06** (37.99%) | **34.70** (55.88%) | **58.62** (56.82%) | **10.78** (4.97%) | 7.47 GB |

To further illustrate the effectiveness of LatentStealth, we conduct qualitative evaluations by visualizing the digital humans generated from both clean and adversarial samples produced by various attack methods. As shown in Fig. 2, LatentStealth leads to noticeable degradation in pose and shape estimation accuracy, resulting in significant deviations in the synthesized digital humans. These visual results clearly highlight the adverse impact of LatentStealth on the EHPS models' ability to accurately reconstruct human poses (Please refer to Appendix C for more results).

## 5.3 ABLATION STUDY

**Effects of Different Noise Injection Strategies.** To further assess LatentStealth, we compare three alternative noise injection strategies within its framework: **A)** adding random noise in the pixel space and updating it iteratively via EHPS test; **B)** injecting random noise into the latent space after VAE encoding, then resampling and decoding to produce adversarial examples; **C)** injecting random noise into the VAE latent space, followed by resampling, decoding, and iterative updates via EHPS queries. We evaluate these strategies on SMPLer-X using the UBody dataset and report GPU memory usage. As shown in Table 2, LatentStealth achieves strong attack performance with only 7.47–12.83 GB of memory, demonstrating its effectiveness and efficiency without requiring extensive resources.

Table 3: Performance comparison of different adversarial attack methods on state-of-the-art EHPS models on the 3DPW dataset. The error growth rates are marked in gray. The maximum error and maximum error growth rate on each setting are **highlighted** underlined.

| Model | Attack | MPJPE (Body) ↓ (mm) | PA MPJPE (Body) ↓ (mm) |
|---|---|---|---|
| | Clean | 75.01 | 50.57 |
| SMPLer-X-H | FGSM | 84.40 (12.52%) | 56.04 (10.82%) |
| | PGD | 88.70 (18.25%) | 57.17 (13.05%) |
| | ACA | 77.30 (3.05%) | 52.47 (3.76%) |
| | DiffAttack | 80.46 (7.27%) | 54.26 (7.30%) |
| | TBA | 88.43 (17.89%) | 59.39 (17.44%) |
| | LatentStealth (Ours) | **103.45** (37.91%) | **65.40** (29.33%) |
| | Clean | 75.85 | 50.67 |
| SMPLer-X-L | FGSM | 85.05 (12.13%) | 56.20 (10.91%) |
| | PGD | 90.16 (18.86%) | 57.30 (12.12%) |
| | ACA | 78.53 (3.53%) | 53.02 (4.64%) |
| | DiffAttack | 78.70 (3.76%) | 53.25 (5.09%) |
| | TBA | 91.63 (20.80%) | 60.54 (19.48%) |
| | LatentStealth (Ours) | **104.53** (37.81%) | **64.94** (28.16%) |
| | Clean | 79.46 | 52.62 |
| SMPLer-X-B | FGSM | 94.04 (18.35%) | 60.97 (15.87%) |
| | PGD | 94.45 (18.86%) | 59.00 (12.12%) |
| | ACA | 82.68 (4.05%) | 55.60 (5.66%) |
| | DiffAttack | 82.93 (4.37%) | 56.31 (7.01%) |
| | TBA | 97.50 (22.70%) | 63.55 (20.77%) |
| | LatentStealth (Ours) | **117.97** (48.46%) | **73.94** (40.52%) |
| | Clean | 82.67 | 56.65 |
| SMPLer-X-S | FGSM | 100.05 (21.02%) | 65.32 (15.30%) |
| | PGD | 98.65 (19.33%) | 63.19 (11.54%) |
| | ACA | 85.52 (3.45%) | 58.86 (3.90%) |
| | DiffAttack | 85.61 (3.56%) | 58.97 (4.10%) |
| | TBA | 108.15 (30.82%) | 69.63 (22.91%) |
| | LatentStealth (Ours) | **117.78** (42.47%) | **74.74** (31.93%) |
| | Clean | 94.32 | 63.87 |
| OSX | FGSM | 102.09 (8.24%) | 68.13 (6.67%) |
| | PGD | 99.58 (5.58%) | 66.01 (3.35%) |
| | ACA | 98.05 (3.95%) | 66.39 (3.95%) |
| | DiffAttack | 97.78 (3.67%) | 66.10 (3.49%) |
| | TBA | 97.33 (3.19%) | 65.32 (3.49%) |
| | LatentStealth (Ours) | **116.89** (23.93%) | **75.34** (17.96%) |
| | Clean | 100.66 | 67.93 |
| Hand4Whole | FGSM | 100.66 (0.00%) | 67.93 (0.00%) |
| | PGD | 100.66 (0.00%) | 67.93 (0.00%) |
| | ACA | 100.70 (0.04%) | 67.95 (0.03%) |
| | DiffAttack | 100.54 (-0.12%) | 67.68 (-0.37%) |
| | TBA | 115.07 (14.32%) | 74.18 (9.20%) |
| | LatentStealth (Ours) | **128.72** (27.88%) | **80.10** (17.92%) |

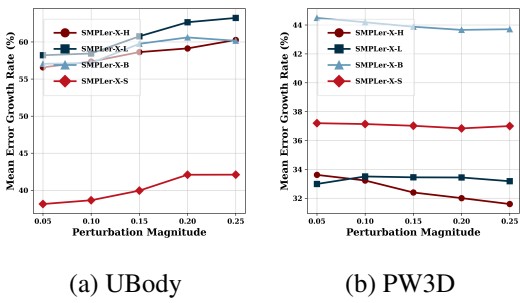

(a) UBody  (b) PW3D

Figure 3: Performance of different perturbation magnitude $\eta$.

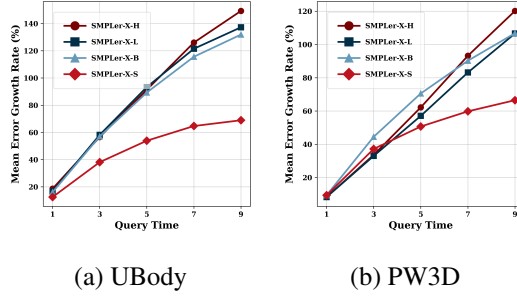

(a) UBody  (b) PW3D

Figure 4: Performance of different test times $t$.

**Effects of Different Perturbation Magnitude $\eta$.** To examine the effect of perturbation magnitude $\eta$ on EHPS models, we evaluate SMPLer-X and report mean error growth rates on the UBody and PW3D datasets (task-specific results are in Appendix C). As shown in Fig. 3, different values of

$\eta$ yield similar attack effectiveness. This indicates that the model's sensitivity to latent-space perturbations is concentrated along specific directions, rather than determined solely by magnitude. Consequently, increasing $\eta$ does not substantially enhance attack performance.

**Effects of Different Test Times $t$.** Similarly, we conduct an ablation study on the number of tests $t$ to the EHPS model, to evaluate its effect on attack performance. Due to space limits, we report only mean error growth rates (task-specific results are in Appendix C). As shown in Fig. 4, $t$ is positively correlated with error growth, indicating that more tests enhance attack effectiveness. When $t = 9$, the mean error growth rates for SMPLer-X-B, SMPLer-X-L, and SMPLer-X-H all exceed 100%. However, excessive tests raise computational cost and reduce efficiency. Although LatentStealth may not perform optimally at $t = 3$, a low-cost attack remains attractive to adversaries.

## 6 CONCLUSION

This paper proposes a novel unnoticeable attack method, termed **LatentStealth**. The proposed method leverages the structured and continuous properties of the latent space in VAE. Specifically, input samples are first projected into the latent space, perturbed with low-magnitude noise, and then decoded back to image space to generate adversarial variants that remain close to the original inputs. The perturbation pattern is iteratively optimized under strict query-budget constraints, with refinement guided by feedback from the EHPS model to ensure computational efficiency. To further balance attack effectiveness and visual imperceptibility, the multi-task loss is employed to jointly optimize adversarial success and perceptual similarity. Experiments show that LatentStealth achieves high attack success, strong imperceptibility, and low computational cost. Beyond EHPS models, this work highlights broader vulnerabilities in EHPS systems and raises awareness in the research community.

## 7 THE USE OF LARGE LANGUAGE MODELS

In preparing this manuscript, we used GPT-5 solely for language editing, including grammar correction and stylistic refinement, to enhance textual clarity and readability. The model was not involved in research ideation, experimental design, data analysis, or result interpretation. All conceptual contributions, technical developments, and scientific conclusions are solely attributable to the authors.

## 8 ETHICS STATEMENT

This work examines the security vulnerabilities of EHPS models through an unnoticable adversarial attack method, **LatentStealth**. The aim is not to promote misuse but to expose weaknesses in digital human generation systems and raise awareness of associated risks. All experiments are conducted on publicly available datasets (3DPW and UBody) that contain no personally identifiable information, with strict adherence to dataset licenses and ethical guidelines. We recognize the potential for negative consequences if adversarial methods are misused, such as compromising security-sensitive applications or producing inappropriate digital human behaviors (Please refer to the Appendix D). To mitigate these risks, our contributions are presented solely for academic purposes, with the explicit goal of improving the robustness and trustworthiness of EHPS technologies. We strongly discourage any harmful, discriminatory, or deceptive use of our method.

## 9 REPRODUCIBILITY STATEMENT

The main paper provides a comprehensive description of the proposed **LatentStealth** method, including the latent-space perturbation mechanism and optimization procedure in Section 4.2. Hyperparameter configurations, evaluation metrics, datasets, and experimental results are reported in Section 5, while additional implementation details and ablation studies are presented in Appendix C to facilitate reproducibility.

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

## A  RESEARCH MOTIVATION & SIGNIFICANCE

Expressive Human Pose and Shape Estimation (EHPS) from monocular images or videos is a pivotal task in digital human generation, with applications encompassing live streaming, virtual reality, and gaming. EHPS is fundamental to creating realistic and interactive digital humans capable of accurately simulating human movements and expressions in real time. However, these systems are highly susceptible to adversarial attacks, which pose substantial risks. For instance, in digital human live streaming, such attacks could cause an avatar's head to appear severed or clothing to fall off, leading to severe incidents involving horror or pornography. In virtual reality environments, such attacks can be exploited by malicious actors to manipulate avatars or even orchestrate harmful scenarios, including acts of virtual terrorism. These potential threats underscore the urgent need for enhanced robustness and security in EHPS systems to mitigate real-world consequences. While this paper investigates adversarial attacks, its primary aim is to raise community awareness regarding this critical issue.

Table 4: Robustness evaluation of the state-of-the-art EHPS model SMPLer-X-H on the UBody dataset under adversarial perturbations from four types of random noise and a central white patch.

| Attack | PA MPVPE ↓ *(mm)* | | | MPVPE ↓ *(mm)* | | | PA MPJPE ↓ *(mm)* | |
|---|---|---|---|---|---|---|---|---|
| | All | Hands | Face | All | Hands | Face | Body | Hands |
| Clean | 24.64 | 8.47 | 2.24 | 41.22 | 30.88 | 16.62 | 29.29 | 8.64 |
| Random Noise ↑ | 24.66 | 8.47 | 2.24 | 41.26 | 30.88 | 16.62 | 29.31 | 8.64 |
| Random Noise ↑↑ | 24.86 | 8.44 | 2.24 | 41.98 | 31.06 | 16.76 | 29.57 | 8.61 |
| Random Noise ↑↑↑ | 25.34 | 8.43 | 2.25 | 44.29 | 31.44 | 17.07 | 30.22 | 8.59 |
| Random Noise ↑↑↑↑ | 25.90 | 8.42 | 2.27 | 47.19 | 31.81 | 17.52 | 30.99 | 8.58 |
| Patch | 24.63 | 8.48 | 2.24 | 41.20 | 30.86 | 16.61 | 29.28 | 8.65 |
| LatentStealth (Ours) | 43.94 | 9.90 | 2.80 | 88.60 | 48.69 | 28.38 | 50.48 | 10.08 |

To evaluate the robustness of the EHPS model, we attack the state-of-the-art SMPLer-X-H using adversarial samples generated from four types of random noise and a central white patch on the UBody dataset. As shown in Table 4, the performance metrics under these attacks deviate negligibly ($\leq 1\%$) from those on clean samples, indicating that ordinary noise does not meaningfully degrade EHPS model performance due to robust generalization from large, diverse training data. In contrast, our proposed LatentStealth successfully compromises SMPLer-X-H, revealing significant vulnerabilities to strategically crafted adversarial samples.

## B  TECHNICAL DETAILS

**Datasets.** We evaluate our method on two benchmark datasets: 3DPW Von Marcard et al. (2018) and UBody Lin et al. (2023). 3DPW comprises in-the-wild RGB video sequences annotated with

3D SMPL poses, capturing diverse human motions in both indoor and outdoor settings using RGB cameras and inertial measurement units (IMUs). UBody is a large-scale dataset for upper-body mesh recovery, serving as a bridge between full-body reconstruction and real-world applications. It contains 1.05 million high-resolution images from 15 real-world scenarios, annotated with 2D whole-body keypoints, person and hand bounding boxes, and SMPL-X parameters.

**EHPS Models.** To assess the effectiveness of LatentStealth, we adopt several state-of-the-art EHPS models. In particular, we utilize Hand4Whole Moon et al. (2022), OSX Lin et al. (2023), and SMPLer-X Cai et al. (2024) in our experiments. SMPLer-X provides four pre-trained foundation models, collectively referred to as "SMPLer-X-M", where M denotes the size of the Vision Transformer (ViT) backbone Alexey (2020), categorized as S, B, L, H.

We select Hand4Whole, OSX, and SMPLer-X to represent the three principal EHPS architectural paradigms. Hand4Whole is a classic multi-stage pipeline excelling in wrist and finger pose estimation, while OSX is the first single-stage Vision Transformer architecture, with high scalability. SMPLer-X utilizes a ViT-Huge backbone trained on tens of millions of samples, emphasizing infrastructure simplification and generalist-to-specialist adaptation. These models are chosen for their public availability, coverage of key paradigms, and wide adoption in the community, ensuring controlled and comparable attack experiments. Models like ExPose Choutas et al. (2020) and FrankMocap Rong et al. (2020), as multi-stage pipeline extensions similar to Hand4Whole, were excluded due to redundancy.

**Baseline and Evaluation Metrics.** Currently, only one adversarial attack has been specifically designed for EHPS. To rigorously evaluate the effectiveness of LatentStealth, we compare it with several state-of-the-art adversarial attack methods in computer vision. Specifically, we assess classical attack algorithms, FGSM Goodfellow et al. (2014) and PGD Madry et al. (2017), alongside two diffusion-based approaches, ACA Chen et al. (2023) and DiffAttack Chen et al. (2024b), as well as the first EHPS-specific attack, TBA Li et al. (2025). For performance evaluation, we report the Mean Per Joint Position Error (MPJPE) and Mean Per-Vertex Position Error (MPVPE), which quantify the accuracy of 3D joint locations and mesh vertex positions, respectively, following prior EHPS benchmarks Moon et al. (2022); Lin et al. (2023); Cai et al. (2024). Additionally, we include PA-MPJPE and PA-MPVPE, which further align rotation and scale. All metrics are measured in millimeters (mm).

**Implementation Details.** The experiments are conducted on an Ubuntu 20.04.6 LTS system using the PyTorch framework, a single NVIDIA L20 GPU, and 45 GB of RAM. The VAE model employed is *stabilityai/sd-vae-ft-mse* from Hugging Face [2]. The maximum perturbation magnitude is set to $\eta = 0.05$ with a step size of 0.01, and an $l_\infty$ constraint was enforced to ensure that changes in individual pixel values did not exceed 0.05. For baseline methods involving iterative optimization, the number of iterations is fixed at 20. For LatentStealth, the number of queries is set to $t = 3$, and the balancing hyperparameter is fixed at $\lambda = 0.5$.

**Algorithmic Flow.** To provide a clearer understanding of the LatentStealth pipeline, we present a detailed the step of generating adversarial samples; see Algorithm 1.

## C   ADDITIONAL EXPERIMENTS

**Effects of Different Perturbation Magnitude $\eta$.** To gain a clearer understanding of the impact of varying perturbation magnitudes $\eta$ on the EHPS model, we conduct experiments using the SMPLer-X model and report the task-specific results for all tasks on the UBody and PW3D datasets, respectively. As illustrated in Tables. 5 and 6, different perturbation magnitudes $\eta$ result in comparable attack effectiveness. This phenomenon may be attributed to the model's sensitivity to latent-space perturbations being concentrated along specific directions rather than being strictly dependent on the perturbation magnitude. As a result, increasing the magnitude does not substantially improve attack performance.

**Effects of Different Query Times $t$.** Similarly, we conduct an ablation study on the number of queries made to the EHPS model API, denoted as $t$, to examine its impact on the performance of adversarial attacks. We also report the task-specific results for all tasks on the UBody and PW3D

---

[2]An open source platform. https://huggingface.co/stabilityai/sd-vae-ft-mse

---

**Algorithm 1** Generation of Adversarial Sample

---

**Input:** Original Image $x$, Random Gaussian Noise $\epsilon_1$, $\epsilon_2$
**Output:** Adversarial Sample $\hat{x}$
**Setting:** Encoder of VAE $\mathcal{E}_\phi$, Decoder of VAE $\mathcal{G}_\theta$, EHPS model $\mathcal{P}$, Mean value $\mu_\phi$, Variance $\sigma_\phi^2$, Number of iterative queries $N$, Step size $\upsilon$, Noise magnitude $\eta$, Learning rate $\zeta$, Latent representation $z$, Multi-task loss function $\mathcal{L}$

1: Loading parameters $\phi$, $\theta$ of VAE
2: $\mu_\phi, \sigma_\phi^2 = \mathcal{E}_\phi(x)$
3: $z = \mu_\phi(x) + \sigma_\phi(x) \odot \epsilon_1$
4: $\hat{z} = z + \eta \times \epsilon_2$
5: $\Delta x = \mathcal{G}_\theta(\hat{z}) - x$
6: $\alpha, \beta, \gamma = P(x)$
7: **for** $i = 1$ to $N$ **do**
8:    $\hat{x} = x + \Delta x$
9:    $\hat{\alpha}, \hat{\beta}, \hat{\gamma} = P(\hat{x})$
10:   $\mathcal{L} = \mathcal{L}(\hat{\alpha}, \hat{\beta}, \hat{\gamma}, \hat{x}, \alpha, \beta, \gamma, x)$
11:   **Update**
12:   $\Delta x = \Delta x - \zeta \nabla_{\Delta x} \mathcal{L}$
13: **end for**
14: **return** $\hat{x} = \Delta x + x$

---

Table 5: Performance comparison of different perturbation magnitude $\eta$ on state-of-the-art EHPS models on the UBody dataset. The error growth rates are marked in gray. The maximum error and maximum error growth rate on each setting are **highlighted** underlined.

| Model | $\eta$ | PA MPVPE ↓ (mm) | | | MPVPE ↓ (mm) | | | PA MPJPE ↓ (mm) | |
|---|---|---|---|---|---|---|---|---|---|
| | | All | Hands | Face | All | Hands | Face | Body | Hands |
| SMPLer-X-H | Clean | 24.64 | 8.47 | 2.24 | 41.22 | 30.88 | 16.62 | 29.29 | 8.64 |
| | 0.05 | 43.94 (78.33%) | 9.90 (16.88%) | 2.80 (25.00%) | 88.60 (114.94%) | 48.69 (57.69%) | 28.38 (70.76%) | 50.48 (72.35%) | 10.08 (16.67%) |
| | 0.10 | 44.46 (80.44%) | 9.94 (17.36%) | 2.79 (24.55%) | 89.22 (116.45%) | 49.44 (60.10%) | 28.03 (68.65%) | 51.03 (74.22%) | 10.11 (17.01%) |
| | 0.15 | 44.73 (81.53%) | 10.02 (18.30%) | **2.82** (25.89%) | 90.74 (120.14%) | 49.62 (60.69%) | 28.02 (68.59%) | 51.46 (75.69%) | 10.21 (18.17%) |
| | 0.20 | 44.51 (80.64%) | **10.13** (19.60%) | 2.80 (25.00%) | 90.76 (120.18%) | 49.76 (61.14%) | **28.54** (71.72%) | 51.38 (75.42%) | **10.31** (19.33%) |
| | 0.25 | **45.43** (84.38%) | 9.98 (17.83%) | 2.81 (25.45%) | **92.81** (125.16%) | **49.98** (61.85%) | 28.20 (69.68%) | **52.79** (80.23%) | 10.16 (17.59%) |
| SMPLer-X-L | Clean | 25.66 | 8.98 | 2.39 | 43.26 | 33.11 | 17.54 | 30.14 | 9.16 |
| | 0.05 | 47.86 (86.52%) | 9.83 (9.47%) | 2.92 (22.18%) | 91.46 (114.42%) | 52.17 (57.57%) | 33.01 (88.20%) | 54.50 (80.82%) | 10.03 (9.50%) |
| | 0.10 | 47.91 (86.71%) | 9.76 (8.69%) | 2.91 (21.76%) | 91.99 (112.64%) | 51.89 (56.72%) | 33.43 (90.59%) | 54.73 (81.59%) | 9.97 (8.84%) |
| | 0.15 | 48.45 (88.82%) | 9.82 (9.35%) | 2.92 (22.18%) | 94.75 (119.02%) | 51.88 (56.69%) | 34.34 (95.78%) | 55.69 (84.77%) | 10.03 (9.50%) |
| | 0.20 | **49.47** (92.79%) | 9.87 (9.91%) | **2.95** (23.43%) | 95.91 (121.71%) | 52.79 (59.44%) | 34.34 (95.78%) | 56.76 (88.32%) | 10.07 (9.93%) |
| | 0.25 | 49.39 (92.48%) | **9.95** (10.80%) | 2.94 (23.01%) | **95.99** (121.89%) | **52.98** (60.01%) | **34.84** (98.63%) | **56.79** (88.42%) | **10.14** (10.70%) |
| SMPLer-X-B | Clean | 28.95 | 9.72 | 2.60 | 50.75 | 38.02 | 19.82 | 33.43 | 9.91 |
| | 0.05 | 53.38 (84.39%) | 10.61 (9.16%) | 3.23 (24.23%) | 106.97 (110.78%) | 57.10 (50.18%) | 36.21 (82.69%) | 62.12 (85.82%) | 10.82 (9.18%) |
| | 0.10 | 53.74 (85.63%) | 10.66 (9.67%) | 3.26 (25.38%) | 106.27 (109.40%) | 56.53 (48.68%) | 35.96 (81.43%) | 62.68 (87.50%) | 10.87 (9.69%) |
| | 0.15 | 54.42 (87.98%) | 10.69 (9.98%) | 3.25 (25.00%) | 110.59 (117.91%) | 57.06 (50.08%) | **36.81** (85.72%) | 64.13 (91.83%) | 10.85 (9.49%) |
| | 0.20 | **54.86** (89.50%) | **10.80** (11.11%) | **3.29** (26.54%) | 110.25 (117.24%) | **57.42** (51.03%) | 36.46 (83.96%) | **65.09** (94.71%) | **10.98** (10.80%) |
| | 0.25 | 54.83 (89.40%) | 10.75 (10.60%) | 3.28 (26.15%) | **111.13** (118.98%) | 56.91 (49.68%) | 36.14 (82.34%) | 64.77 (93.75%) | 10.94 (10.39%) |
| SMPLer-X-S | Clean | 32.28 | 10.05 | 2.83 | 57.23 | 42.80 | 22.26 | 37.38 | 10.27 |
| | 0.05 | 50.16 (55.39%) | 10.53 (4.78%) | 3.33 (17.67%) | 98.24 (71.66%) | 59.06 (37.99%) | 34.70 (55.88%) | 58.62 (56.82%) | 10.78 (4.97%) |
| | 0.10 | 50.28 (55.76%) | 10.60 (5.47%) | 3.32 (17.31%) | 98.31 (71.78%) | 59.02 (37.90%) | 35.40 (59.03%) | 58.52 (56.55%) | 10.82 (5.36%) |
| | 0.15 | 50.69 (57.03%) | 10.60 (5.47%) | 3.35 (18.37%) | 100.39 (75.41%) | 59.54 (39.11%) | 35.84 (61.01%) | 58.99 (57.81%) | 10.82 (5.36%) |
| | 0.20 | **51.24** (58.74%) | **10.72** (6.67%) | 3.38 (19.43%) | 102.18 (78.54%) | 59.70 (39.49%) | **37.11** (66.71%) | **60.07** (60.70%) | **10.93** (6.43%) |
| | 0.25 | 50.90 (57.68%) | **10.72** (6.67%) | **3.40** (20.14%) | **103.03** (80.03%) | **59.98** (40.14%) | 36.88 (65.68%) | 59.87 (60.17%) | 10.92 (6.33%) |

datasets, respectively. As shown in Tables 7 and 8, the number of queries $t$ is positively correlated with the mean error growth rate, suggesting that increasing the number of queries enhances attack effectiveness. Notably, the error growth rate exceeds 100% on both MPVPE and PA MPJPE multiple estimation metrics, and even reaches a maximum of 255.45%. However, an excessive number of queries raises the attacker's cost and may alert the platform or API owner. Therefore, although such attacks may not yield optimal performance, conducting them within a strict query budget enables sustained adversarial behavior.

We adopt four standard image quality evaluation metrics, e.g., PSNR (Peak Signal-to-Noise Ratio), SSIM (Structural Similarity Index) Wang et al. (2004), LPIPS (Learned Perceptual Image Patch Similarity) Zhang et al. (2018), and FID (Fréchet Inception Distance) Heusel et al. (2017), to assess the imperceptibility of the adversarial examples generated by our method. As presented in Table 9, our method achieves the best results in PSNR and FID, and second-best performance in SSIM and LPIPS. Furthermore, we conduct additional qualitative experiments by visualizing the generated digital humans from both clean and adversarial samples generated by various attack methods across

Table 6: Performance comparison of different perturbation magnitude $\eta$ on state-of-the-art EHPS models on the PW3D dataset. The error growth rates are marked in gray. The maximum error and maximum error growth rate on each setting are **highlighted** underlined.

| Model | $\eta$ | MPJPE (Body) ↓ (mm) | PA MPJPE (Body) ↓ (mm) |
|---|---|---|---|
| | Clean | 75.01 | 50.57 |
| SMPLer-X-H | 0.05 | **103.45** (37.91%) | 65.40 (29.33%) |
| | 0.10 | 102.85 (37.12%) | **65.41** (29.35%) |
| | 0.15 | 101.91 (35.86%) | 65.20 (28.93%) |
| | 0.20 | 101.48 (35.29%) | 65.10 (28.73%) |
| | 0.25 | 101.12 (34.81%) | 64.93 (28.40%) |
| | Clean | 75.85 | 50.67 |
| SMPLer-X-L | 0.05 | 104.53 (37.81%) | 64.94 (28.16%) |
| | 0.10 | **104.81** (38.18%) | 65.28 (28.83%) |
| | 0.15 | 104.64 (37.96%) | 65.34 (28.95%) |
| | 0.20 | 104.55 (37.84%) | **65.39** (29.05%) |
| | 0.25 | 104.17 (37.34%) | 65.38 (29.03%) |
| | Clean | 79.46 | 52.62 |
| SMPLer-X-B | 0.05 | **117.97** (48.46%) | **73.94** (40.52%) |
| | 0.10 | 117.58 (47.97%) | 73.88 (40.40%) |
| | 0.15 | 117.53 (47.91%) | 73.59 (39.85%) |
| | 0.20 | 117.42 (47.77%) | 73.43 (39.55%) |
| | 0.25 | 117.60 (48.00%) | 73.36 (39.41%) |
| | Clean | 82.67 | 56.65 |
| SMPLer-X-S | 0.05 | 117.78 (42.47%) | **74.74** (31.93%) |
| | 0.10 | 118.07 (42.82%) | 74.47 (31.46%) |
| | 0.15 | 118.16 (42.93%) | 74.27 (31.10%) |
| | 0.20 | 118.13 (42.89%) | 74.09 (30.79%) |
| | 0.25 | **118.32** (43.12%) | 74.14 (30.87%) |

Table 7: Performance comparison of different query times $t$ on state-of-the-art EHPS models on the PW3D dataset. The error growth rates are marked in gray. The maximum error and maximum error growth rate on each setting are **highlighted** underlined.

| Model | Attack | MPJPE (Body) ↓ (mm) | PA MPJPE (Body) ↓ (mm) |
|---|---|---|---|
| | Clean | 75.01 | 50.57 |
| SMPLer-X-H | 1 | 82.28 (9.69%) | 54.03 (6.84%) |
| | 3 | 103.45 (37.91%) | 65.40 (29.33%) |
| | 5 | 127.04 (69.36%) | 78.41 (55.05%) |
| | 7 | 153.67 (104.87%) | 91.81 (81.55%) |
| | 9 | **177.11** (136.12%) | **103.29** (104.25%) |
| | Clean | 75.85 | 50.67 |
| SMPLer-X-L | 1 | 82.97 (9.39%) | 54.29 (7.14%) |
| | 3 | 104.53 (37.81%) | 64.94 (28.16%) |
| | 5 | 124.51 (64.15%) | 76.01 (50.01%) |
| | 7 | 146.68 (93.38%) | 87.63 (72.94%) |
| | 9 | **166.96** (120.12%) | **97.83** (93.07%) |
| | Clean | 79.46 | 52.62 |
| SMPLer-X-B | 1 | 87.34 (9.92%) | 57.28 (8.86%) |
| | 3 | 117.97 (48.46%) | 73.94 (40.52%) |
| | 5 | 140.52 (76.84%) | 86.45 (64.29%) |
| | 7 | 157.20 (97.84%) | 96.19 (82.80%) |
| | 9 | **171.32** (115.61%) | **104.14** (97.91%) |
| | Clean | 82.67 | 56.65 |
| SMPLer-X-S | 1 | 91.62 (10.83%) | 61.12 (7.89%) |
| | 3 | 117.78 (42.47%) | 74.74 (31.93%) |
| | 5 | 131.04 (58.51%) | 80.92 (42.84%) |
| | 7 | 139.82 (69.13%) | 85.24 (50.47%) |
| | 9 | **146.17** (76.81%) | **88.47** (56.17%) |

Table 8: Performance comparison of different query times $t$ on state-of-the-art EHPS models on the UBody dataset. The error growth rates are marked in gray. The maximum error and maximum error growth rate on each setting are **highlighted** underlined.

| Model | $t$ | PA MPVPE ↓ (mm) | | | MPVPE ↓ (mm) | | | PA MPJPE ↓ (mm) | |
|---|---|---|---|---|---|---|---|---|---|
| | | All | Hands | Face | All | Hands | Face | Body | Hands |
| | Clean | 24.64 | 8.47 | 2.24 | 41.22 | 30.88 | 16.62 | 29.29 | 8.64 |
| SMPLer-X-H | 1 | 30.74 (24.76%) | 8.99 (6.14%) | 2.41 (7.59%) | 58.90 (42.89%) | 36.80 (19.17%) | 20.00 (20.34%) | 35.67 (21.78%) | 9.19 (6.37%) |
| | 3 | 43.94 (78.33%) | 9.90 (16.88%) | 2.80 (25.00%) | 88.60 (114.94%) | 48.69 (57.69%) | 28.38 (70.76%) | 50.48 (72.35%) | 10.08 (16.67%) |
| | 5 | 55.14 (123.78%) | 10.27 (21.25%) | 3.22 (43.75%) | 115.86 (181.08%) | 57.41 (85.91%) | 39.33 (136.64%) | 63.62 (117.21%) | 10.45 (20.95%) |
| | 7 | 64.32 (161.04%) | 10.69 (26.21%) | 3.59 (60.27%) | 144.96 (251.67%) | 66.02 (113.80%) | 51.29 (208.60%) | 76.34 (160.64%) | 10.89 (26.04%) |
| | 9 | **70.83** (187.46%) | **11.01** (29.99%) | **3.89** (73.66%) | **161.56** (291.95%) | **73.72** (138.73%) | **59.29** (256.74%) | **83.76** (185.97%) | **11.21** (29.75%) |
| | Clean | 25.66 | 8.98 | 2.39 | 43.26 | 33.11 | 17.54 | 30.14 | 9.16 |
| SMPLer-X-L | 1 | 32.52 (26.73%) | 9.05 (0.78%) | 2.49 (4.18%) | 59.09 (36.59%) | 39.72 (19.96%) | 21.18 (20.75%) | 37.36 (23.95%) | 9.23 (0.76%) |
| | 3 | 47.86 (86.52%) | 9.83 (9.47%) | 2.92 (22.18%) | 91.46 (114.42%) | 52.17 (57.57%) | 33.01 (88.20%) | 54.50 (80.82%) | 10.03 (9.50%) |
| | 5 | 59.03 (130.05%) | 10.34 (15.14%) | 3.31 (38.49%) | 120.70 (179.01%) | 61.16 (84.72%) | 44.91 (156.04%) | 68.73 (128.04%) | 10.53 (14.96%) |
| | 7 | 65.75 (156.24%) | 10.62 (18.26%) | 3.71 (55.23%) | 146.06 (237.63%) | 69.93 (111.21%) | 55.43 (216.02%) | 78.09 (159.09%) | 10.82 (18.12%) |
| | 9 | **71.06** (176.93%) | **11.01** (22.61%) | **3.91** (63.60%) | **157.47** (264.01%) | **75.02** (126.58%) | **60.80** (246.64%) | **82.93** (175.15%) | **11.23** (22.60%) |
| | Clean | 28.95 | 9.72 | 2.60 | 50.75 | 38.02 | 19.82 | 33.43 | 9.91 |
| SMPLer-X-B | 1 | 36.04 (24.49%) | 9.87 (1.54%) | 2.83 (8.85%) | 68.59 (35.15%) | 43.18 (13.57%) | 23.92 (20.69%) | 41.09 (22.91%) | 10.08 (1.72%) |
| | 3 | 53.38 (84.39%) | 10.61 (9.16%) | 3.23 (24.23%) | 106.97 (110.78%) | 57.10 (50.18%) | 36.21 (82.69%) | 62.12 (85.82%) | 10.82 (9.18%) |
| | 5 | 64.52 (122.87%) | 11.12 (14.40%) | 3.60 (38.46%) | 135.60 (167.19%) | 68.92 (81.27%) | 48.92 (146.82%) | 76.61 (129.17%) | 11.33 (14.33%) |
| | 7 | 71.25 (146.11%) | 11.56 (18.93%) | 3.96 (52.31%) | 161.01 (217.26%) | 79.57 (109.28%) | 61.61 (210.85%) | 84.00 (151.27%) | 11.77 (18.77%) |
| | 9 | **74.96** (158.93%) | **11.69** (20.27%) | **4.25** (63.46%) | **176.13** (247.05%) | **85.22** (124.15%) | **70.45** (255.45%) | **88.87** (165.84%) | **11.90** (20.08%) |
| | Clean | 32.28 | 10.05 | 2.83 | 57.23 | 42.80 | 22.26 | 37.38 | 10.27 |
| SMPLer-X-S | 1 | 38.71 (19.92%) | 10.11 (0.60%) | 3.03 (7.07%) | 70.78 (23.68%) | 48.63 (13.62%) | 25.55 (14.78%) | 44.76 (19.74%) | 10.34 (0.68%) |
| | 3 | 50.16 (55.39%) | 10.53 (4.78%) | 3.33 (17.67%) | 98.24 (71.66%) | 59.06 (37.99%) | 34.70 (55.88%) | 58.62 (56.82%) | 10.78 (4.97%) |
| | 5 | 54.82 (69.83%) | 10.82 (7.66%) | 3.55 (25.44%) | 114.22 (99.58%) | 65.40 (52.80%) | 44.03 (97.80%) | 63.69 (70.39%) | 11.06 (7.69%) |
| | 7 | 57.40 (77.82%) | 10.96 (9.05%) | 3.70 (30.74%) | 125.21 (118.78%) | 70.09 (63.76%) | 51.48 (131.27%) | 66.17 (77.02%) | 11.20 (9.06%) |
| | 9 | **58.03** (79.77%) | **11.01** (9.55%) | **3.75** (32.51%) | **130.08** (127.29%) | **71.89** (67.97%) | **54.82** (146.27%) | **66.76** (78.60%) | **11.25** (9.54%) |

various scenarios to substantiate our findings. As shown in Figures. 5 and 6, LatentStealth causes a marked degradation in human pose and shape estimation accuracy, leading to evident distortions in the synthesized digital humans, and in some instances, rendering the generated poses entirely implausible. These results clearly demonstrate the adverse impact of LatentStealth on the ability of EHPS models to accurately capture human pose.

## D  SOCIAL IMPACT

**Positive Social Impacts.** Although our work centers on adversarial attacks, its primary objective is to uncover vulnerabilities in digital human generation models and raise community awareness of the

Table 9: Quantitative comparison on UBody. The best and the second best results are **highlighted** and underlined.

| Method | PSNR ↑ | SSIM ↑ | LPIPS ↓ | FID ↓ | $L_2$ ↓ | $L_\infty$ ↓ |
|---|---|---|---|---|---|---|
| FGSM | 46.69 | 0.98 | 0.13 | 2.64 | 3.56 | **0.01** |
| PGD | 40.79 | 0.92 | 0.23 | 7.24 | 7.01 | 0.03 |
| ACA | 29.01 | **0.99** | **0.01** | 11.26 | 31.76 | 0.10 |
| DiffAttack | 40.63 | 0.92 | 0.19 | 7.43 | 7.15 | 0.05 |
| TBA | 13.83 | 0.62 | 0.54 | 62.04 | 157.52 | 0.48 |
| LatentStealth (Ours) | **47.15** | 0.98 | 0.10 | **1.89** | **3.37** | **0.01** |

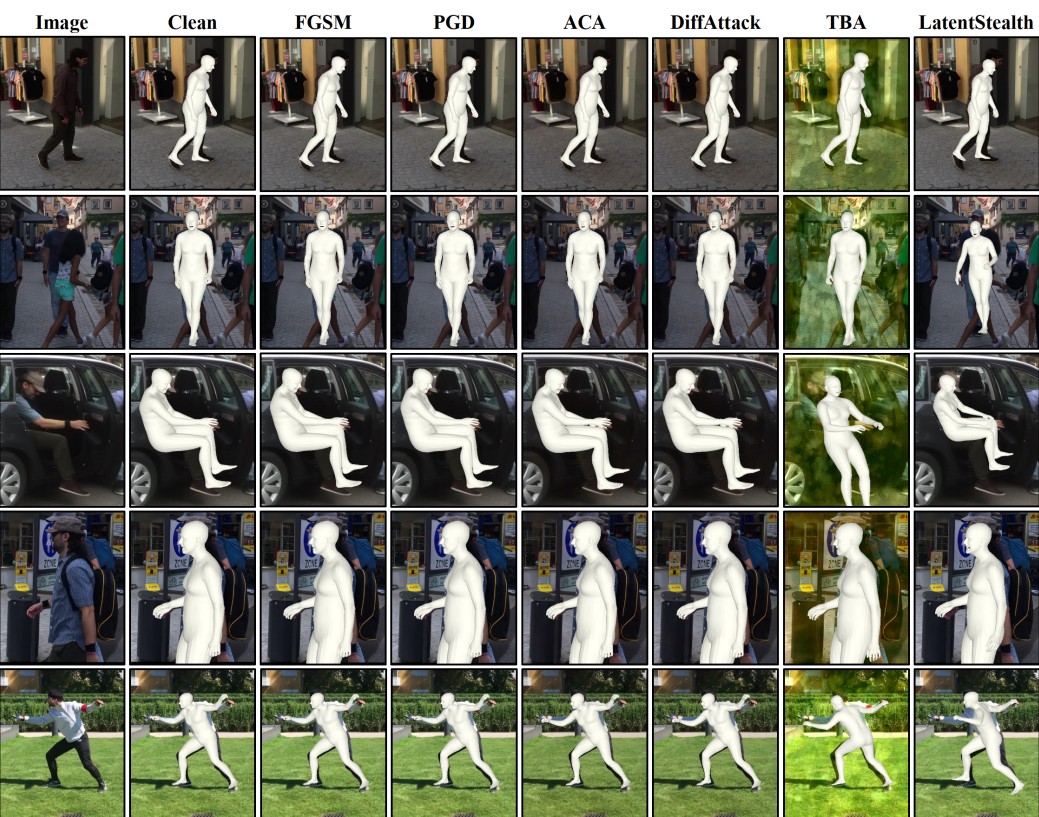

Figure 5: Visualizing various adversarial samples for digital human generation on PW3D.

associated security risks. We seek to alert developers to these issues and promote the development of robust and secure digital human synthesis systems.

**Negative Social Impacts.** The experimental results validate the proposed method's strong applicability and attack effectiveness. However, its potential misuse by malicious actors to compromise security-sensitive applications raises serious concerns about the robustness of digital human generation technologies. For instance, in digital human live streaming, such attacks could cause an avatar's head to appear severed or clothing to fall off, leading to severe incidents involving horror or pornography.

## E    LIMITATIONS

While this work proposes an effective and imperceptible attack scheme with strong real-world applicability, it also has limitations. Specifically, our evaluation focuses exclusively on attack efficacy, without considering the potential influence of standard input pre-processing defenses. This omission implies a potentially unrealistic assumption—that the API provider or model owner does not apply

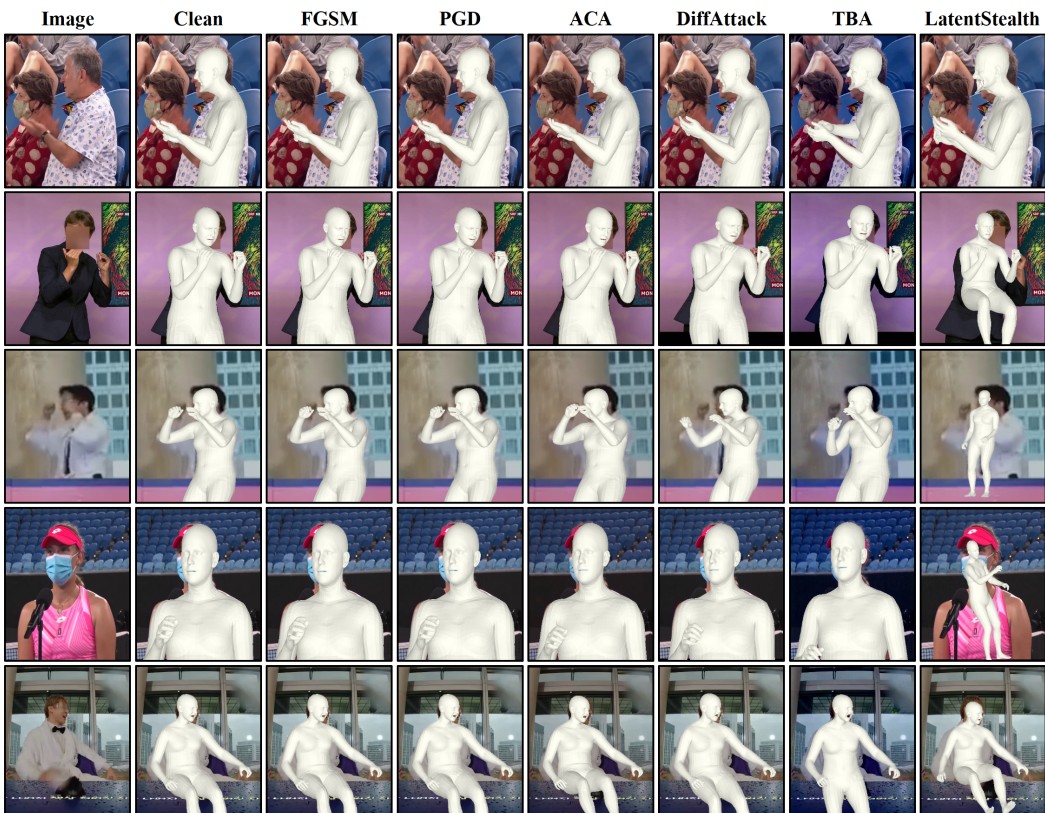

Figure 6: Visualizing various adversarial samples for digital human generation on UBody.

any input transformations or defense techniques (e.g., image compression, smoothing filters, JPEG encoding, or feature denoising) before inference. However, such defenses are often integrated into secure machine learning pipelines and may significantly degrade or nullify the adversarial perturbations generated by our method. Therefore, future research should systematically examine the effect of the proposed attack under various pre-processing defenses and adaptive countermeasures to more accurately assess its practical threat in real-world deployments.

