# OpenReview forum: "LatentStealth: Unnoticeable and Efficient Adversarial Attacks on Expressive Human Pose and Shape Estimation"
_ICLR.cc/2026/Conference — ICLR 2026 Conference Withdrawn Submission_

### Official Review · Reviewer_6DHA · 2025-10-26

**Soundness:** 2
**Presentation:** 3
**Contribution:** 2
**Rating:** 2
**Confidence:** 3

**Summary:**

The paper introduces LatentStealth, a novel adversarial attack method for expressive human pose and shape estimation (EHPS) models. It highlights that while existing EHPS models reduce estimation errors, they often neglect security vulnerabilities. Current attacks produce noticeable perturbations, limiting practicality. In contrast, LatentStealth generates imperceptible adversarial examples by manipulating the latent space of a Variational Autoencoder (VAE) using low-magnitude Gaussian noise and iterative optimization. Experiments on the 3DPW and UBody datasets show significant improvements in attack performance with low computational costs.

**Strengths:**

1. This paper is easy to follow.
2. The research problem is interesting. The exploration of adversarial attacks on EHPS models addresses a timely and relevant issue in digital human modeling.
3. Demonstrates significant increases in pose estimation errors.

**Weaknesses:**

1. While the paper claims that optimization is performed in latent space, in practice only the initial noise in the first step is generated there. The subsequent optimization takes place in pixel space, following a procedure similar to Projected Gradient Descent (PGD), according to Eq.18-22. Moreover, as stated in Line 353, the first step uses a fixed value of η (0.05) rather than selecting it according to Equation 13. This design choice raises concerns about the effectiveness of the initial step and, consequently, casts doubt on the authors’ assertion of achieving substantially better performance than PGD.
2. In Figure 2, the images for the FGSM, PGD, and PCA methods are the same as the clean image, but according to Table 1, they should be different.
3. In the introduction, the authors mention a potential application scenario involving the generation of disturbing scenes, such as a broken neck. However, the method they actually use adopts an untargeted attack objective—maximizing the overall error. It would be more convincing if the authors could evaluate targeted attacks, for instance, focusing on specific regions such as the neck.
4. The paper states that the perturbations produced by TBA are visually perceptible and fail to pass human inspection. However, they did not conduct comparison experiments on the visual qualities of TBA and their own method.
5. Lack of transferability experiments. Does the method  be transferable to black-box models?

Small error:
1. The citation format is incorrect; the name should be in brackets. Please refer to ICLR's template

**Questions:**

See weakness

---

### Official Review · Reviewer_VQXZ · 2025-10-28

**Soundness:** 2
**Presentation:** 3
**Contribution:** 2
**Rating:** 2
**Confidence:** 4

**Summary:**

LatentStealth introduces an adversarial attack for expressive human pose & shape estimation (EHPS). It perturbs a pretrained VAE’s latent code to create imperceptible seeds, then refines them in pixel space with a multi-task loss that maximizes pose/shape errors while constraining visual distortion.

**Strengths:**

1. LatentStealth is a black-box attack method where the attacker can only access the model’s output under a strict query budget.

2. The validation covers two benchmarks (3DPW, UBody) and across multiple EHPS backbones/variants

**Weaknesses:**

1. I found the pipeline of this work is too similar to the work of Li et al. (Pattern Recognition 2026); the core components such as latent space noise injection, noise optimization, and enhancement are almost identical. Although the authors replace the first stage with a VAE architecture, I believe the essence of the methods is the same. This makes me think the paper lacks real novelty.

2. The Output-access assumption may be unrealistic. The loss is defined on full EHPS parameters $P(x)=(\alpha,\beta,\gamma)$; many production APIs expose only meshes, 2D/3D keypoints, or rendered frames. The paper does not test such constrained-output settings.

3. The whole attack pipeline over relies on a specific VAE. The attack is tied to stabilityai/sd-vae-ft-mse; there’s no study of robustness to swapping the generative prior or to distribution shift between the VAE’s training data and EHPS inputs.

4. The Query-efficiency not benchmarked against modern black-box attacks. Comparisons include FGSM/PGD and diffusion-based baselines, but omit standard query-based black-box methods (e.g., NES/Square) under matched query budgets, leaving the efficiency claim under-substantiated.

REF: Li Z, Jin Y, Shen F, et al. Unveiling hidden vulnerabilities in digital human generation via adversarial attacks[J]. Pattern Recognition, 2026.

**Questions:**

See the weakness part of my review.

---

### Official Review · Reviewer_m8fQ · 2025-10-30

**Soundness:** 2
**Presentation:** 2
**Contribution:** 2
**Rating:** 4
**Confidence:** 3

**Summary:**

This paper introduces LatentStealth, the first imperceptible adversarial attack designed for expressive human pose and shape estimation (EHPS) models. Instead of perturbing pixel space, the method injects low-magnitude noise into the latent space of a pretrained VAE and refines perturbations via a multi-task loss, ensuring both attack effectiveness and visual stealthiness under strict query budgets. Experiments on datasets 3DPW and UBody demonstrate substantial error increases across state-of-the-art EHPS models, while maintaining imperceptibility and efficiency. This work exposes critical security risks in digital human modeling and provides a novel attack paradigm.

**Strengths:**

[1] The paper introduces the first method that perturbs the latent space of a pretrained VAE to attack expressive human pose and shape estimation. This is a simple but powerful idea: perturbations added in latent space generate adversarial images that are visually indistinguishable from the original in pixel space, making the attack more stealthy.
[2] The proposed multi-task loss elegantly trades off adversarial strength with visual fidelity, offering a practical blueprint for adversarial optimization under realistic constraints.

**Weaknesses:**

[1] The paper should more precisely state its novelty (“the first systematic application of latent-space attacks to EHPS”) and directly compare/contrast with existing latent-space adversarial work. The authors must highlight what is genuinely new in algorithmic design, loss formulation, or query-efficient optimization.
[2] The paper references live streaming, but experiments are on static images. The authors should evaluate temporal coherence and runtime/query cost on continuous frames or restrict claims to static images.
[3] Current experiments use a pre-trained VAE and a few EHPS backbones, some older. The authors should compare against more recent EHPS models and test the attack with other generative priors to demonstrate whether the method is broadly applicable.
[4] The paper injects Gaussian noise into latent space as an initialization and then refines in pixel space. However, practical threats may require targeted manipulations (forcing specific pose/shape changes). The authors should show how to extend their approach to targeted attacks and evaluate the trade-off between stealthiness and success rate. The current study primarily demonstrates untargeted degradation.
[5] Lack of defense discussion or experiments. Security papers should at least discuss plausible defenses and provide preliminary evaluations.

**Questions:**

Your abstract claims “the first systematic application of latent-space attacks to EHPS.” What is precisely novel relative to existing latent-space adversarial work? Please specify concrete additions in (a) algorithmic design, (b) loss formulation, and (c) query-efficient optimization, and provide side-by-side comparisons.

Your text references live streaming, but experiments are on static images. Also, the evaluation uses a pre-trained VAE and a few (some older) EHPS backbones. Can you (a) validate temporal coherence and per-frame runtime/query budgets on short video clips, and (b) test on newer EHPS models and alternative generative priors to demonstrate breadth? If not feasible now, will you limit claims to static images and a specific prior?

Your current pipeline injects Gaussian noise in latent space and refines in pixel space, mainly showing untargeted degradation. Can you extend to targeted manipulations (e.g., specific pose/shape changes) and analyze the stealthiness–success trade-off, and discuss and preliminarily evaluate defenses?

---

### Official Review · Reviewer_BuJd · 2025-10-31

**Soundness:** 2
**Presentation:** 3
**Contribution:** 2
**Rating:** 2
**Confidence:** 5

**Summary:**

This paper proposes a unnoticeable adversarial attack, LatentStealth, against expressive human pose and shape estimation (EHPS) models. The input samples are first projected into the latent space, perturbed with low-magnitude noise, and then decoded back to image space to generate adversarial variants that remain close to the original inputs. The results showed an improvement in estimated pose errors compared to baselines.

**Strengths:**

The paper writing is clear and easy to follow. The authors visualize the results of their attack, which clearly shows that the adversarial examples generated by their method induce significant errors in the estimated human poses.

**Weaknesses:**

1. The motivation of this attack is not very clear. The authors should provide more details about the scenarios in which this attack will be used and explain why it is important to attack human pose and shape estimation.

2. The proposed method lacks novelty. Perturbing the latent space of generative models has been widely used in adversarial attacks for various vision tasks. The authors should explain why this method is especially better for the EHPS models.

3. The comparison with baselines is unfair. The perturbation of the baseline methods is constrained by the l-inf norm, while that of the proposed method is softly constrained. The authors should provide more analysis on how this affects the results, such as comparing the norm of the perturbations generated by different methods, and evaluating the baselines (e.g., C&W attack) with similar perturbation norms.

**Questions:**

1. Which VAE model is better to use in this work? Will it be better to be trained on the same dataset as the EHPS models?

2. What do the adversarial examples look like for different methods? What if using different constraints on the perturbations?

---

### Note · Authors · 2025-11-13

I have read and agree with the venue's withdrawal policy on behalf of myself and my co-authors.